# Conventional measures of intrinsic excitability are poor estimators of neuronal activity under realistic synaptic inputs

**Adrienn Szabó**[1], **Katalin Schlett**[1], **Attila Szücs**[1,2]*

**1** Neuronal Cell Biology Research Group, Eötvös Loránd University, Budapest, Hungary, **2** BioCircuits Institute, University of California San Diego, La Jolla, California, United States of America

* attila.szucs@ttk.elte.hu

## Abstract

Activity-dependent regulation of intrinsic excitability has been shown to greatly contribute to the overall plasticity of neuronal circuits. Such neuroadaptations are commonly investigated in patch clamp experiments using current step stimulation and the resulting input-output functions are analyzed to quantify alterations in intrinsic excitability. However, it is rarely addressed, how such changes translate to the function of neurons when they operate under natural synaptic inputs. Still, it is reasonable to expect that a strong correlation and near pro-portional relationship exist between static firing responses and those evoked by synaptic drive. We challenge this view by performing a high-yield electrophysiological analysis of cultured mouse hippocampal neurons using both standard protocols and simulated synaptic inputs via dynamic clamp. We find that under these conditions the neurons exhibit vastly dif-ferent firing responses with surprisingly weak correlation between static and dynamic firing intensities. These contrasting responses are regulated by two intrinsic K-currents mediated by Kv1 and $K_{ir}$ channels, respectively. Pharmacological manipulation of the K-currents pro-duces differential regulation of the firing output of neurons. Static firing responses are greatly increased in stuttering type neurons under blocking their Kv1 channels, while the synaptic responses of the same neurons are less affected. Pharmacological blocking of $K_{ir}$-channels in delayed firing type neurons, on the other hand, exhibit the opposite effects. Our subse-quent computational model simulations confirm the findings in the electrophysiological experiments and also show that adaptive changes in the kinetic properties of such currents can even produce paradoxical regulation of the firing output.

## Author summary

Most action potentials that neurons emit during their lifetime are produced by a dynamic interplay between the synaptic inputs and the intrinsic biophysical properties of the post-synaptic neuron. Activity-dependent or neuromodulatory changes targeting these intrin-sic properties effectively regulate intrinsic excitability of the neurons and how they integrate synaptic input into firing output. Electrophysiologists mostly employ current

**Data Availability Statement:** The raw model data data and electrophysiological recordings were uploaded to the Figshare repository (https://figshare.com/browse) and are accessible via 10.

6084/m9.figshare.16536171 and 10.6084/m9.
figshare.16554003.

**Funding:** This research was supported by the
Hungarian National Brain Research Program under
grant 2017-1.2.1-NKP-2017-00002 to KS; by the
National Research, Development and Innovation
Office of Hungary under grant VEKOP-2.3.3-15-
2016-00007 to KS; by the Hungarian Scientific
Research Foundation under grant ANN-135291 to
ASz; and by the ELTE Thematic Excellence
Programme 2020 (TKP2020-IKA-05) to KS. The
funders had no role in study design, data collection
and analysis, decision to publish, or preparation of
the manuscript.

**Competing interests:** The authors have declared
that no competing interests exist.

step protocols in whole-cell patch clamp experiments to identify such changes in intrinsic
excitability and to estimate the underlying functional consequences. In the present study
we investigate the firing output of hundreds of hippocampal neurons under standard cur-
rent step stimulation and when they are bombarded by simulated synaptic inputs via
dynamic clamp. Our experiments show that firing intensity values in the two scenarios
exhibit a surprisingly low correlation, hence, static firing responses yield poor predictive
power to estimate firing responses under synaptic inputs. We also show in electrophysio-
logical experiments and computer simulations that two voltage-dependent K-currents
mediated by Kv1 and $K_{ir}$-channels in stuttering and delayed firing type neurons, respec-
tively, play a key role in regulating these differential firing responses.

## Introduction

In addition to the well-known forms of synaptic plasticity, intrinsic properties of neurons are
regulated by activity-dependent mechanisms. Such modifications, mainly associated with spe-
cific voltage-activated membrane currents, greatly contribute to the overall functional plasticity
ity of neuronal networks, because they directly impact how synaptic inputs are translated to
action potential output [1,2]. As a prime example of such intrinsic plasticity in hippocampal
circuits, Kv1 channels mediating the D-type K-current in parvalbumin-expressing interneu-
rons are downregulated after LTP-induction via Schaffer-collateral stimulation [3]. The corre-
sponding changes in intrinsic excitability of the basket cells facilitate their recruitment in the
network activity at gamma-frequencies. Similar form of Kv1 channel mediated long-term
potentiation of intrinsic excitability (LTP-IE) has been found in CA3 neurons evoked solely by
somatic electrical stimulation [4]. LTP-IE often involves the downregulation of various K-
channels [4–8], but additional ion channel targets have been also identified [9,10]. All these
findings indicate that intrinsic adaptations play as important role in cellular mechanisms of
learning as the well-known synaptic forms of long-term plasticity.

Researchers aiming to uncover potential effects of activity-dependent intrinsic adaptations
commonly use standard step current stimulation in whole-cell patch clamp settings. In such
experiments, gradually more depolarizing levels of current are applied to elicit firing responses
and to obtain the neurons' input-output functions. While this approach has been very success-
ful in detecting changes in intrinsic excitability due to activity-dependent plasticity or chronic
neuroadaptations [11–14], we are to recognize that neurons in vivo receive rapidly fluctuating
synaptic conductances [15] rather than stepwise levels of transmembrane current. It is there-
fore reasonable to examine how well the analysis of firing responses under static current sti-
muli can predict functional adaptations of neurons when they operate in their natural synaptic
environment. This is an important question because activity-dependent up- or downregula-
tion of specific voltage-gated currents can greatly alter the operation of neuronal circuits. Yet,
this problem is rarely addressed in electrophysiological studies, partly because it is challenging
to accurately control the synaptic inputs of a neuron during its intrinsic adaptations and to
compare the resulting firing responses to those under static current steps.

Our present study tackles this problem by performing a high-yield comparative analysis of
static excitability and firing responses under simulated synaptic inputs in mouse hippocampal
neurons. We show that these neurons, belonging to physiologically different phenotypes,
exhibit vastly different firing responses under static stimulation vs. synaptic drive. These con-
trasting responses are mediated by the actions of the D-type K-current and the inward rectify-
ing K-current that exhibit differential impact on the regulation of firing responses. Findings

from our electrophysiological observations are reinforced by model simulations of the firing responses of the biophysically diverse neurons. Our modeling also identifies specific biophysical parameters that facilitate the differential regulatory effects. Paradoxically, potential changes in the kinetics of the D-current can even result in upregulation of static firing responses while reducing the synaptic responses in the same neurons.

## Results

### Cultured hippocampal neurons exhibit diverse physiological properties

Hippocampal neurons in primary dissociated cell cultures exhibit a high level of diversity in their voltage responses under current step stimulation 12–14 days after plating. Currents step protocols, widely used by electrophysiologists, serve as a very effective experimental technique to extract a number of informative physiological parameters [16]. Importantly, physiological properties determined in such experiments also provide a solid basis for classification of cell types that are differentiated according to their intrinsic biophysical properties [17–19]. Hence, we performed a standardized current step stimulation protocol on each neuron analyzed in this study. Although we observed a high variety of voltage responses of the cultured hippocampal neurons, we were able to assign those into 3 main categories, as shown in Fig 1. In particular, regular firing type neurons exhibited robust spiking under moderate depolarizing current levels, voltage sag under negative currents and low rheobase (the threshold current level where spiking initiated) that was mostly below +100 pA (Fig 1A, 1D and 1E). Spike count, as a function of the injected current was fairly continuous and monotonously increasing. Neurons assigned to the second group and referred to as delayed firing type neurons, displayed prominent inward rectification under negative current steps and a slow voltage ramp preceding the first spike emission (Fig 1B, 1F and 1G). The third type of neurons exhibited low membrane resistance, outward rectification, voltage sag and intermittent spiking responses even at current levels above +100 pA (Fig 1C). Often, only a single spike was elicited near rheobase and an irregular firing response was produced at more depolarized current levels. We often observed a discontinuity in the spike count vs. current relationship for these type of neurons. This behavior is commonly referred to as stuttering [3,20], therefore we used this term to label neurons assigned to the third group. While the single spike response often seen in the stuttering neurons indicated a low overall excitability of such cells, this spike was emitted very rapidly following the onset of the positive current step (Fig 1C and 1I). Hence, first spike latency serves as one of the discriminating physiological parameters suitable to separate delayed firing and stuttering type cells. The most important features that were used to identify such stuttering type neurons were the strong outward rectification, marked spike afterhyperpolarization, discontinuous I-O relationship, single spike response near rheobase, voltage sag and short spike latency.

Altogether we recorded the voltage responses of 414 neurons that yielded the densely populated 2-dimensional representations of their selected physiological parameters shown in Fig 1J, 1K and 1L. Based on these maps, the three populations of cells, corresponding to the regular, delayed and stuttering types, largely overlap, indicating no clear distinction among the 3 categories. Depending on the selected pairs of physiological parameters, we find a variable degree of separation among the 3 groups. As an example, the input resistance vs. resting membrane potential or the latency$_{1.3}$ vs. input resistance plots clearly separate the delayed firing and stuttering type neurons (Fig 1J and 1K). Separation of these two types is less obvious when we examine the total spike count vs. rheobase relationship (Fig 1L). In agreement with our subjective assessment, the analysis of the physiological parameters identifies these two types as being the most distinguishable. On the other hand, regular firing type neurons (see gray symbols)

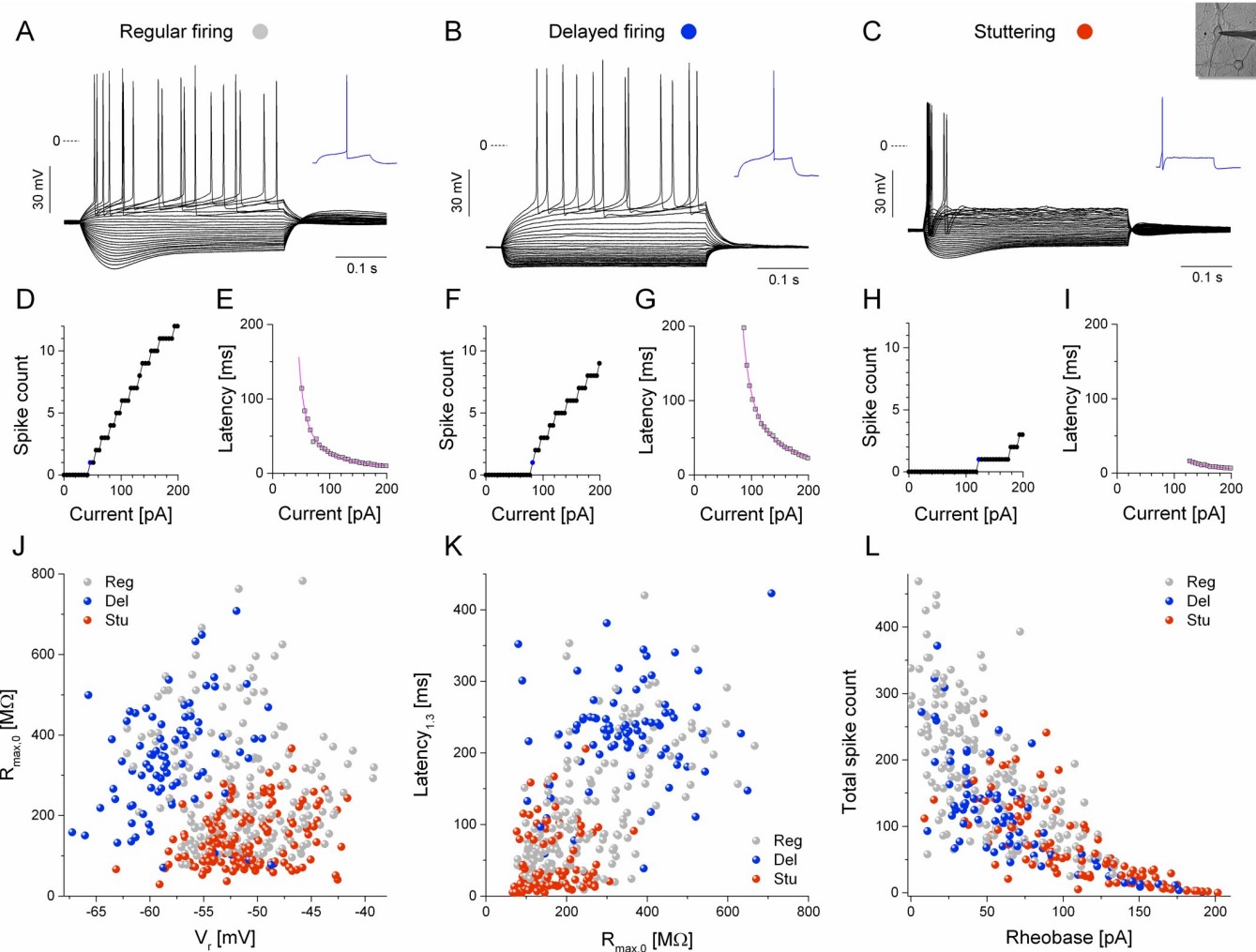

**Fig 1. Cultured hippocampal neurons exhibit diverse physiological properties.** (A-C) Voltage responses of 3 types of neurons evoked by current step stimulation. Blue traces show the first spike responses just above rheobase levels (also indicated by blue symbols in the I-O curves). Spike count vs. current relationship for the regular, delayed firing and stuttering neurons are shown in D, F and H, respectively. The first spike latency as a function of the injected current is plotted in E, G, I. The magenta lines are fitted Belehradek functions. (J-L) Cells assigned to one of the three phenotypes are visualized in scatter plots of various pairs of extracted physiological parameters. (J) The input membrane resistance plotted against the resting membrane potential; (K) first spike latency at 1.3-times of the rheobase current level plotted against the input resistance; (L) total (cumulative) spike count vs. rheobase. Colors indicate the subjective neuronal phenotypes, gray: regular firing, blue: delayed firing, red: stuttering type cells.

are the most variable as their parameters overlap with those of the stuttering and delayed firing type cells.

## Excitability measures greatly differ among neuron types receiving static vs. dynamic stimuli

The marked differences in the voltage responses of the delayed and stuttering neurons indicate differences in their intrinsic biophysical properties such as the expression patterns of their specific voltage-activated currents. Indeed, the prominent inward rectification and delayed onset of firing are known characteristics of the action of inward rectifying K-channels [21,22]. Stuttering type neurons, on the other hand, display a reduction in the observed membrane resistance when applying positive step currents (outward rectification). Additionally, at suprathreshold current levels,

action potentials are tailed by strong afterhyperpolarization. These features are consistent with the action of D-type K-currents mediated by Kv1 channels [20,23]. Our earlier model simulations and dynamic clamp study revealed differential regulation of firing responses under the manipulation of the neurons' multiple voltage-gated currents [24,25] including the two K-currents associated with the delayed firing and stuttering behavior. Based on these, we hypothesize that static and dynamic firing responses become increasingly divergent when cultured hippocampal neurons of this biophysically diverse population are exposed to such inputs.

The hippocampal cell culture, therefore, serves as a prime biological system to test this idea. Our next set of experiments were aimed at comparing the firing responses of a high number of neurons under static current stimulation and under simulated synaptic inputs via dynamic clamp. The two protocols yielded firing responses that allowed the construction of the corresponding I-O functions. We note that a small percentage of neurons, typically stuttering ones, did not reach firing threshold even at the highest stimulus intensities (I = +200 pA and $g_{AMPA}$ = 25 nS, respectively), hence, such experiments resulted in missing data for the rheobase and/or threshold AMPA-conductance and yielded 0 for the total spike count. Yet, these two selected values for maximal stimulation intensity were found to be applicable for the great majority of cells. Fig 2 illustrates the behavior of one representative delayed firing and one stuttering type neurons under such protocols. Here, the delayed type neuron fired healthily under step currents emitting a gradually increasing number of spikes (Fig 2A and 2B). Conversely, and in agreement with our expectation [24,25], the same neuron fired with less intensity when receiving the simulated synaptic inputs (Fig 2C and 2D). Stuttering neurons, on the other hand, fired sparsely under current steps but vigorously under dynamic clamp (Fig 2E, 2F and 2G, 2H, respectively). Thus, static and dynamic firing responses of these two types of neurons were quite opposite.

The distinction of the firing responses is further demonstrated by plotting the threshold AMPA-conductance as a function of the threshold current level (rheobase) for all the neurons recorded (n = 380). We observed a slight separation of the delayed firing vs. stuttering type neurons in this representation. Indeed, stuttering neurons require a relatively stronger stimulation to initiate firing under current steps than delayed firing type cells, but they start firing at lower AMPA-conductance levels than the other type (Fig 2I). Regular firing type neurons, on the other hand, appear as a dispersed population largely overlapping with the other 2 types (see grey symbols on Fig 2I).

Expectedly, neurons with higher rheobase would also require higher threshold conductance under simulated synaptic inputs and, in general, this is what we observe in the mapping of the firing responses. However, the overall correlation between the two threshold parameters is less than expected, yielding a Spearman-coefficient of 0.704 for all the neurons measured in these experiments. Cumulative (total) spike counts recorded under the 2 types of stimulation serve as more preferable measures of the firing responses, because they take into account all the spikes emitted under the protocols, i.e. responses from dozens of stimulus sweeps. Importantly, such total spike counts recorded under static vs. dynamic stimulation exhibit even lower correlation as shown by the high dispersion of data in Fig 2J (Corr$_S$ = 0.614), further supporting our original hypothesis. We conclude that the rheobase and total spike count parameters calculated from current step responses offer a poor estimation for the neurons' firing intensity under realistic synaptic inputs. This is especially true when low total spike counts, associated with near-threshold stimulus intensities are considered.

## Variation of voltage-gated currents in computational models reproduces physiological diversity

To better explain the role of biophysical properties in shaping the differential responses and experimentally observed physiological properties of hippocampal neurons, we constructed

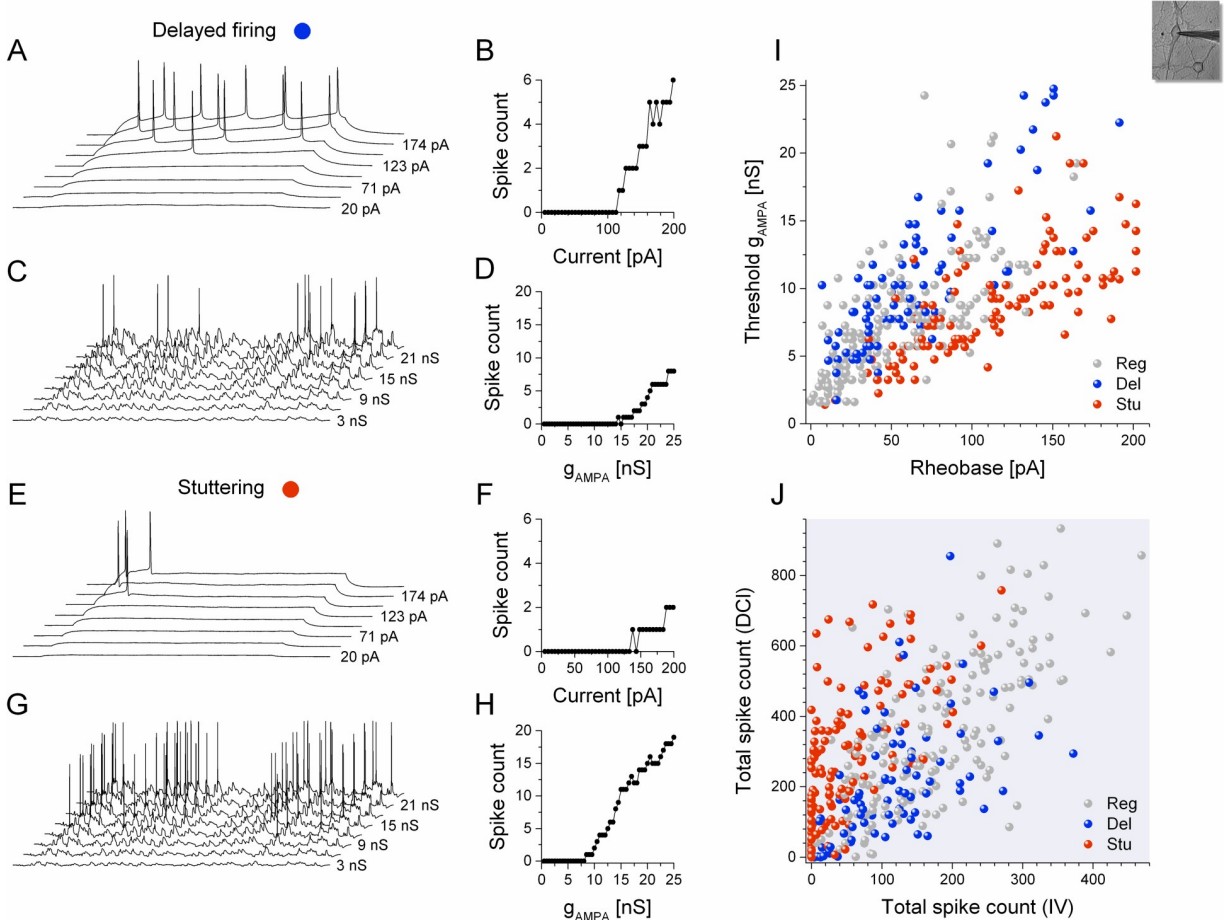

**Fig 2. Firing responses of the hippocampal neurons indicate different degree of excitability under current step stimulation vs. simulated synaptic bombardment.** Voltage traces of a delayed firing type cell and the corresponding input-output relationship are shown under current step protocol in (A) and (B), respectively. (C) Firing response and (D) the spike count vs. AMPA-conductance relationship obtained from the same delayed type neuron under dynamic clamp protocol. Corresponding panels in (E-H) demonstrate the firing responses of a stuttering type cell under identical stimulus conditions. Note that the I-O relationships are very different between the two neuron types (B vs. F and D vs. H). (I) The threshold AMPA-conductance is plotted against the rheobase for all the recorded neurons (n = 380). (J) Total (cumulative) spike count from the dynamic clamp experiments is plotted against the total static spike count (n = 399). Colors indicate the three cell types.

computational models based on three generic phenotypes. The regular firing, delayed firing and stuttering type model neurons were designed to closely match the dynamical behavior of their corresponding biological phenotypes and the physiological parameters measured in current step experiments. As an example, voltage sag, commonly observed both in regular firing and stuttering neurons was reproduced by including the nonspecific, hyperpolarization-activated cation current ($I_h$) in these 2 models (Fig 3A and 3C) [19,25]. In contrast, the inward rectification and delayed onset of spiking (latency) was excellently reproduced by the addition of the inward rectifying K-current ($I_{Kir}$) to the delayed firing type model [25]. Stuttering as observed in the third type of neurons, was facilitated by a potent D-current that activated and inactivated slowly [20,26].

As shown above, cultured hippocampal neurons exhibit a variety of firing responses and physiological parameters that are likely caused by cell-to-cell variations in the strength of their specific voltage-activated currents and morphological properties among others. We replicated, at least partially, such variations in the model implementations by randomly varying the

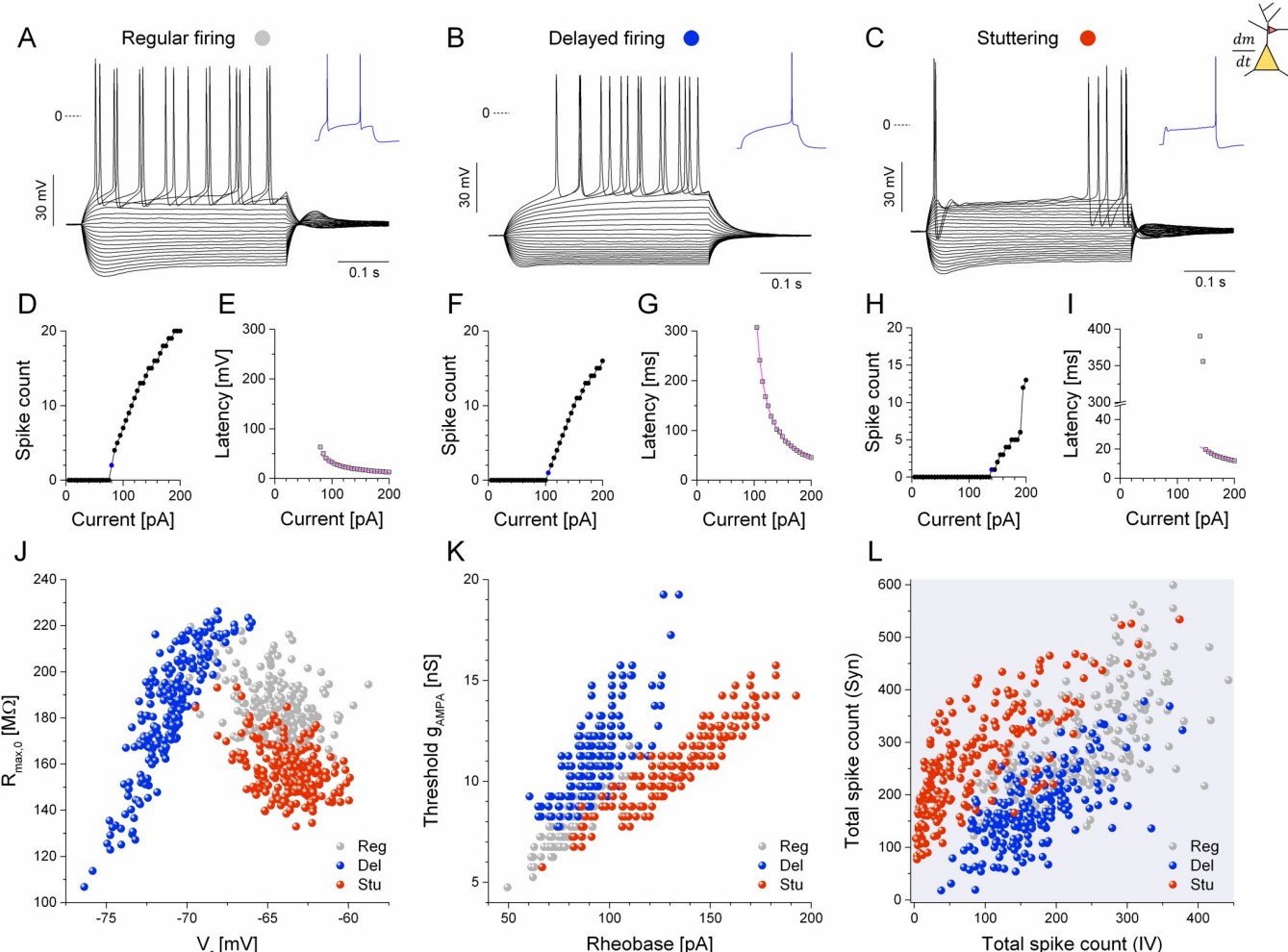

**Fig 3. The model neurons reproduce the firing responses and the physiological diversity of the biological neurons observed in the experiments.** (A, B, C) Voltage responses of the regular, delayed and stuttering type models driven by current step stimulation. Blue traces show the first spike responses just above rheobase levels. (D, F, H) Spike count vs. current plots and (E, G, I) first spike latency plotted as the function of the injected current are shown for the regular, delayed and stuttering type model neurons. Note that the stuttering model exhibits a sudden jump in spike count at the transition to repetitive firing (+195 pA, H). (J) Input membrane resistance vs. resting membrane potential of all model instances. (K) The threshold AMPA-conductance is plotted against the rheobase of the model neurons. (L) Cumulative spike counts from synaptic vs. current step responses are scatter plotted. Data from 600 model instances are shown.

maximal conductances of selected voltage-gated currents, the membrane capacitance and leakage membrane resistance parameters. For each model phenotype we generated 200 instances and subjected those to the two types of stimulation as used with the biological neurons. The analysis of the observable physiological properties and firing responses was then performed identically to that we have used in the electrophysiological experiments.

Examples of the three model phenotypes are shown in Fig 3 as they fire under the action of somatic step current injection (identical to that we used for the biological neurons). Input-output curves (Fig 3D, 3F and 3H) and first spike latency plots (Fig 3E, 3G and 3I) of these model neurons well reproduce the corresponding data from the biological neurons. However, spike latency functions of stuttering type model neurons often exhibit discontinuity (Fig 3I) because the short latency action potentials at the beginning of the step appear at current levels slightly above rheobase. In such model neurons we find a spikelet at the beginning of the step and a full action potential following the voltage ramp (due to slow inactivation of the intrinsic D-

current, Fig 3C, blue trace). The membrane resistance vs. resting $V_m$ plot is qualitatively similar to that of the biological neurons, but the overall spread of points is lower, and the three model phenotypes are better separated than the corresponding biological neurons (compare Fig 3J with Fig 1J). More importantly, the analysis of the firing responses of the model neurons largely reproduces the low degree of correlation between static and dynamic responses of real neurons: the relationship between the threshold AMPA-conductance and the rheobase (compare Fig 3K with Fig 2I) as well as the scatter plot of the total (cumulative) spike counts (Fig 3L vs. Fig 2J) are very similar to what we obtained for the biological neurons. In addition to the good qualitative reproduction of the biological data, we find a low Spearman-correlation calculated for the cumulative spike count data ($Corr_S = 0.531$, n = 600).

Altogether, these observations support the idea that variations in the magnitude of specific voltage-gated membrane currents are, at least partially, responsible for the observed physiological diversity of hippocampal neurons. Besides, we identify potential candidates of membrane currents that play a key role in the differential regulation of firing responses of these neurons, particularly the delayed firing and stuttering type ones.

## Differential effects of blocking the intrinsic D- and $K_{ir}$-currents on firing responses

Assuming the dominant role of the $K_{ir}$- and D-current in shaping the excitability profiles of the hippocampal neurons, one can expect that pharmacological blocking of these specific currents would change their firing properties in a differential manner depending on the type of stimulus they receive. To test this idea, we patched neurons exhibiting clear inward rectification, the hallmark of delayed firing type neurons (see Fig 4A). Next, we recorded their physiological properties and obtained their static and dynamic I-O functions (Fig 4C and 4D). Then, we bath-applied BaCl$_2$ (125 μM) and repeated the same two protocols that allowed us to examine the changes of the neurons' excitability profiles after the blocking of their $K_{ir}$-current (Fig 4B, 4C and 4D) [27,28]. As shown for the representative neuron, blocking of its intrinsic $K_{ir}$-current removed the characteristic inward rectification, clearly increased the membrane resistance (and time constant) and moderately increased the number of spikes emitted during the positive current steps (Fig 4C and 4E). In case of the I-O functions, we found a stronger enhancement of firing during the simulated synaptic inputs than under the static current step protocol (Fig 4D and 4F). This finding is in good agreement with our prior computational modeling results and dynamic clamp experiments when a computer-synthesized $K_{ir}$-current was inserted into the regular firing neurons (as reverse experiment) [24]. Following BaCl$_2$ application, the mean relative increase of total spike counts was +304% in dynamic clamp conditions (n = 15) and +67% when current steps were used (n = 20) (Fig 4G). Clearly, blocking the $K_{ir}$-current regulated the firing responses of the delayed firing type neurons in a significantly different manner depending on the inputs they received.

The same experimental protocol was applied to stuttering neurons when their intrinsic D-current was blocked by 20 μM 4-AP, an effective antagonist of Kv1 channels (Fig 4H–4N) [3,29]. Quite the contrary to what we found with the delayed firing type neurons, the simulated synaptic responses of these cells were barely affected by D-current blocking (Fig 4M), however, their static firing responses increased dramatically (Fig 4L). In fact, these neurons, initially responding with single spikes or irregular bursts under step currents, turned into more regular firing neurons after 4-AP application (Fig 4H and 4I). The differential regulation of the firing is clearly shown by the corresponding I-O functions (compare Fig 4J and 4L with Fig 4K and 4M, respectively; n = 22). In average, total spike counts under static stimulation went up by +1570% and only by +59% under simulated synaptic inputs (Fig 4N, n = 22 for both). The firing reducing effect of the

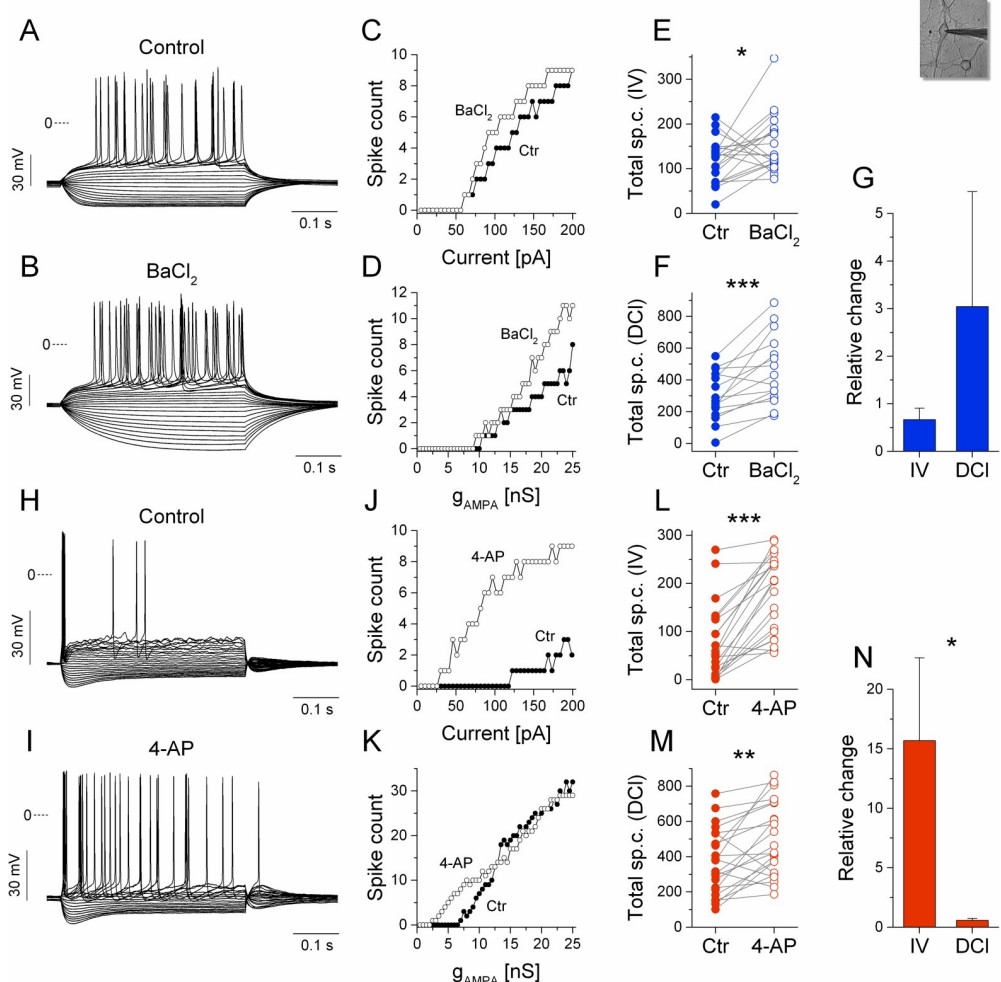

**Fig 4. Pharmacological blocking of the $K_{ir}$- and D-currents produce differential effects in delayed vs. stuttering type neurons.** The voltage responses of a delayed firing type neuron are shown before (A) and after the application of $BaCl_2$ (B). (C) Input-output functions obtained from the current step experiments indicate a moderate increase of the firing under $BaCl_2$. (D) I-O relationship obtained from the dynamic clamp experiments. (E and F) Total spike counts in control and $BaCl_2$-treated cells are shown for the current step vs. DCl experiments, respectively. (G) The average relative change in total spike counts in the current step (IV) and dynamic clamp (DCl) experiments. Panels (H-N) show the same for the stuttering type neurons under the application of 4-AP. The firing of the demonstrated neuron is markedly increased by 4-AP application when the current step stimulation is used (J) but increased to a less degree when the simulated synaptic inputs are used (K). The pooled data (L-M) reveal that spike counts under current step stimulation increase far more than firing under simulated synaptic inputs. (N) Average relative change of total spike counts following 4-AP application in the current step (IV) and dynamic clamp (DCl) experiments.

D-current is the strongest slightly above rheobase when the neurons typically emit single spikes or stutter. Blocking this current has therefore a great impact on the static responses, because single spiking or sparse stuttering is now replaced with regular spiking.

## Manipulations of K-currents in computational models

The pharmacological experiments, described above, suggest that delayed firing and stuttering type neurons represent opposite extremes in the spectrum of cultured hippocampal neurons, as far as their physiological properties are considered. Indeed, the 2- or higher dimensional distributions of the physiological parameters (Fig 1J, 1K and 1L) suggest the same.

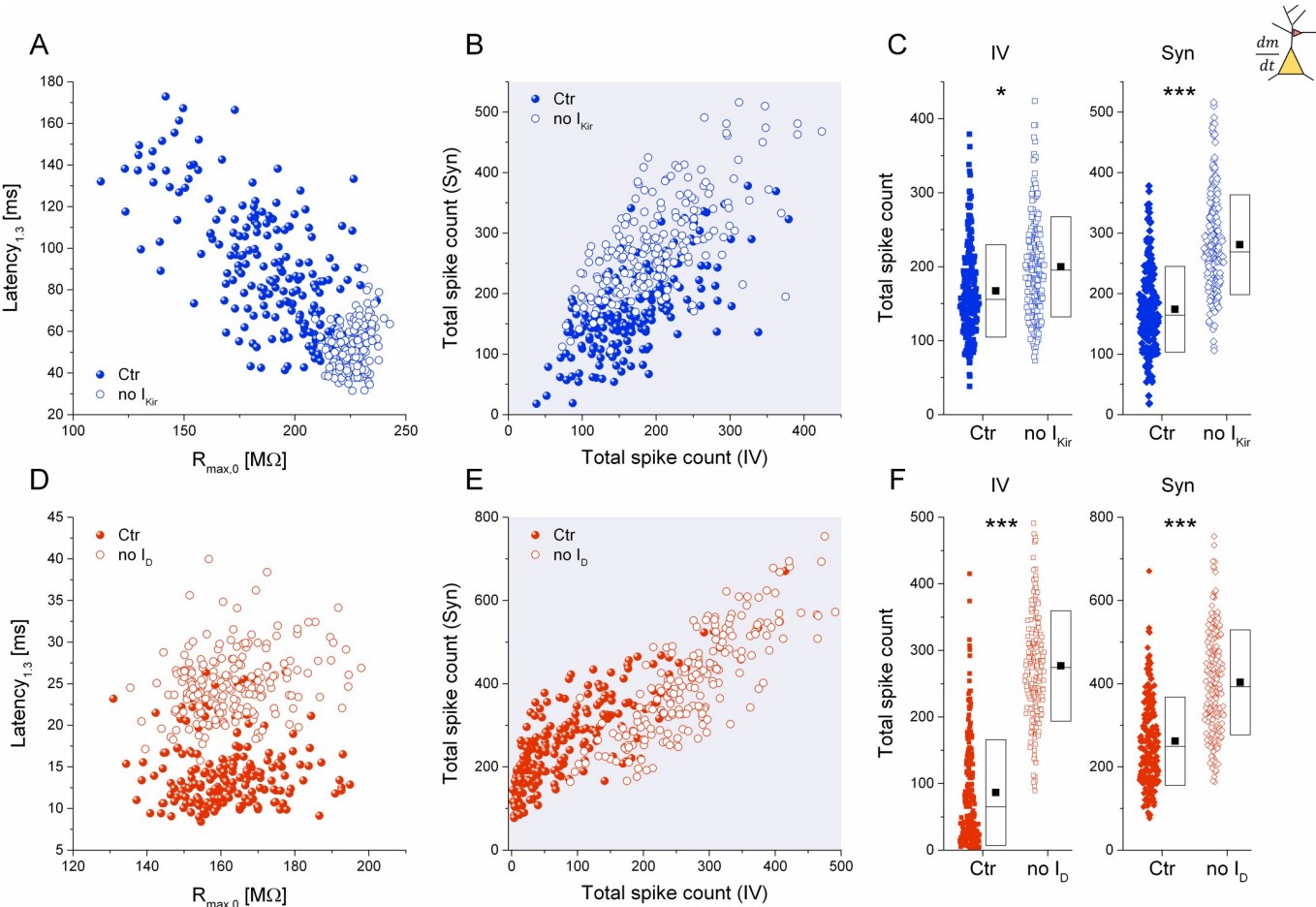

**Fig 5. Removal of two K-currents from the model neurons produce differential effects on their firing output.** (A) First spike latency vs. input resistance plot for the delayed firing type neurons. Non-filled symbols represent the data after the removal of the $K_{ir}$-current. (B) Scatter plot of cumulative spike counts from the same set of model neurons (synaptic vs. current step responses). (C) Pooled cumulative spike counts shift in a differential manner when static vs. dynamic inputs are used. D-F: Same plots demonstrating the effects of D-current removal from the stuttering type model neurons.

Accordingly, selectively blocking the $K_{ir}$- and D-currents in these two cell types would expectedly reduce their separation.

We investigated this possibility in model simulations on the delayed firing and stuttering type neurons before and after the removal of their corresponding voltage-gated K-currents while applying current step stimulation or simulated synaptic bombardment as above (Fig 5). As expected, removal of the $K_{ir}$-current from the delayed firing type model and the D-current from the stuttering one resulted in profound changes in many of their observable physiological parameters. Eliminating the $I_{Kir}$ increased the membrane resistance of the model neurons while reducing their latency parameter (Fig 5A). This action clearly shows the role of the $K_{ir}$-current in prolonging the first spike latency during suprathreshold current injections. While the delaying of the first spike is usually associated with D-currents [3], $I_{Kir}$ is solely responsible for the delayed onset of firing (and ramp potential) in these models indicating the same role of $I_{Kir}$ in the corresponding biological neurons. Somewhat paradoxically, the latency parameter increased after D-current removal in the stuttering models (Fig 5D) indicating that these cells start firing with longer latencies but at lower step current levels compared to those having their intrinsic D-currents intact.

          

The effects of virtual pharmacological manipulations on firing output are shown in Fig 5B and 5E, by plotting spike counts from the dynamic clamp models against those obtained from the current step model. The simulations clearly demonstrate a general shift of the entire population after the removal of the $K_{ir}$- and D-currents. Remarkably, the excitability parameters change in a differential manner, i.e. the effect of $I_{Kir}$-removal is stronger on the synaptic responses than the current step responses (Fig 5B) while the opposite is true for the D-current (Fig 5E). In addition, firing responses of the neurons lacking $K_{ir}$- and D-currents became more uniform, both under current steps and simulated synaptic drive. The scatter/box plots of the total spike counts confirm the differential effects of $K_{ir}$ vs. D-current blocking (Fig 5C and 5F). Taken together, these simulated data are in an excellent agreement with our electrophysiological recordings from real neurons and they confirm the key role of these 2 voltage-gated currents in regulating the firing output.

## Voltage dependence and kinetics of the K-currents

The two voltage-dependent K-currents, playing central role in our study, have different activation properties. Specifically, the $K_{ir}$-current activates at hyperpolarized membrane potentials while the characteristic voltage ramp and delayed onset of firing is caused by the current's deactivation during depolarizing current stimuli. Conversely, the D-current activates with depolarization and exhibits a slow inactivation, too. Can such differences in their properties explain the differential regulation of firing responses under the action of these currents? To examine this possibility, we ran model simulations and manipulated selected biophysical parameters of these currents while measuring the firing responses of the neurons under the two stimulus settings.

The $K_{ir}$-current is characterized by a negative slope sigmoidal steady-state activation curve and the first set of simulations used the midpoint of the sigmoid ($V_{m,1/2}$) as control parameter. We selected a subset of the 200 delayed firing type model neurons, 25 instances, that well represented the the entire population. Then, we performed the static and dynamic stimulus protocols to gain the total spike counts as above. For each model implementation we incremented the $V_{m,1/2}$ parameter from the low -91 mV to -75 mV by +2 mV and obtained the excitability measures in 9 settings. Effectively, we shifted the steady-state activation curve of the $I_{Kir}$ and analyzed the firing responses of the models. Fig 6A shows the summary of such model runs. Here, each string of symbols corresponds to a single model implementation with 9 consecutive values for the $V_{m,1/2}$ parameter. Shifting the activation midpoint to more depolarized values clearly decreases both the static and dynamic spike counts. Indeed, for each model implementation we find nearly parallel trajectories that descend toward the origin of this map. This scatter map is not unlike the original excitability map of the 200 delayed firing neuron models where the maximal conductances of the intrinsic currents, including $I_{Kir}$, were randomly varied (Fig 3L). Hence, the effect of shifting the activation curve of the $K_{ir}$-current is similar to that of changing its maximal conductance.

In addition to varying the voltage-dependence of the activation profile, we investigated the effects of changing the activation time constant of the $K_{ir}$-current. Interestingly, manipulating the kinetics of the $K_{ir}$-current has a minor effect on the excitability measures. We adjusted the maximal time constant of activation from 10 ms to 130 ms in 15 ms steps and obtained similar trajectories as in the previous case. Fig 6B shows that the changes in firing responses under this manipulation were far weaker than the ones under $V_{m,1/2}$ adjustment. Again, we find trajectories descending toward the origin, although their individual length is far less than in Fig 6A. This observation shows that the impact of $I_{Kir}$ in regulating the excitability of the model neurons is fairly consistent in a wide range of activation time constants.

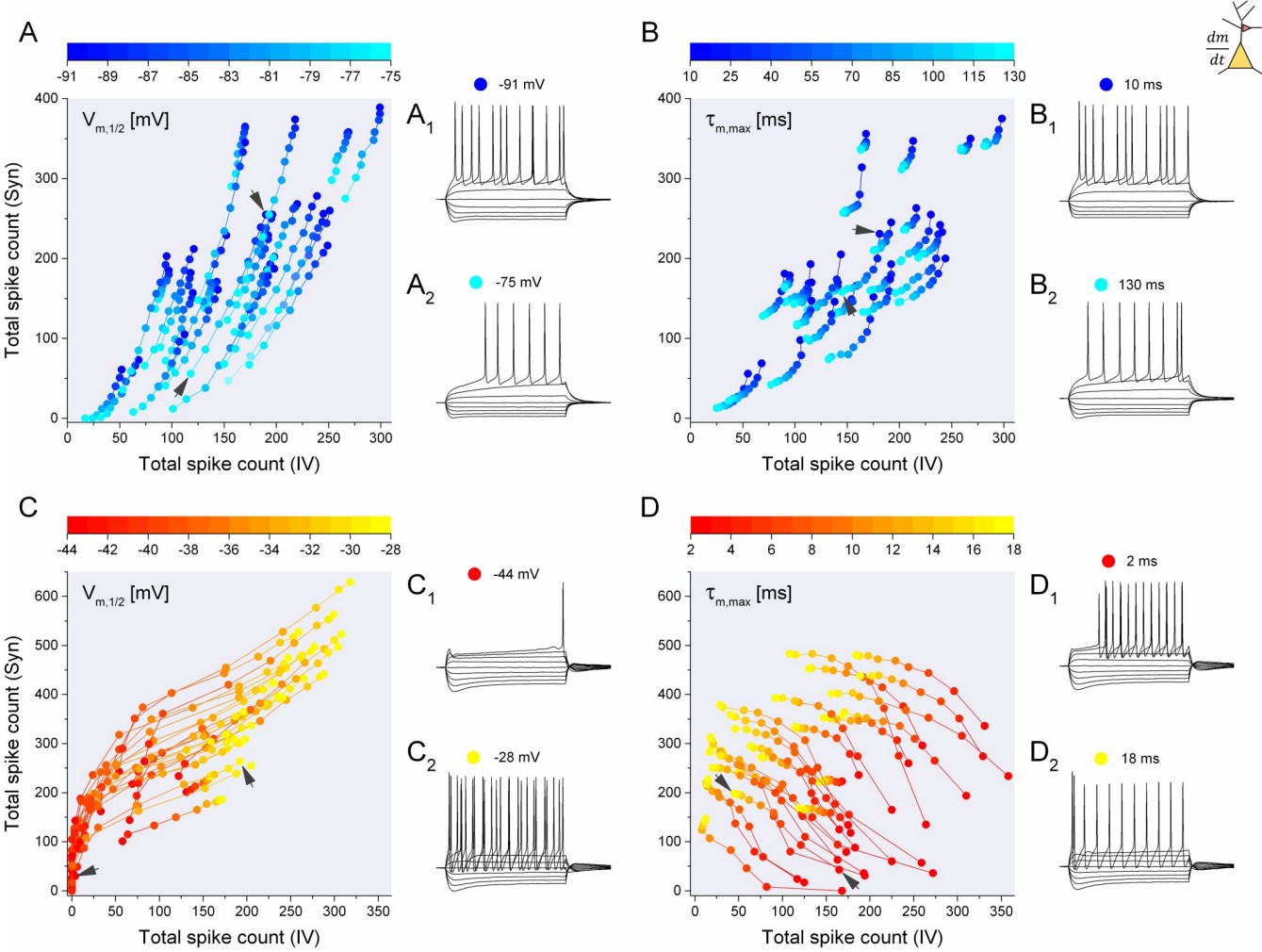

**Fig 6. Manipulations of the voltage dependence and kinetics of the K-currents exert profound effects on the firing responses of the model neurons.** (A) The half-voltage of the steady-state activation curve ($V_{m,1/2}$) of the $K_{ir}$-current is shifted from -91 to -75 mV and the cumulative spike counts of 25 model instances are calculated. Black arrows indicate the model responses shown in $A_1$ and $A_2$. (B) The peak activation time constant of the Kir-current ($\tau_{m,max}$) is shifted from 10 to 130 ms and the cumulative spike counts of the corresponding model responses are plotted. (C) Similar manipulation is performed on the steady-state activation midpoint of the D-current of the stuttering type model (25 instances). (D) Peak activation time constant of the D-current is shifted from 2 to 18 ms. Here, models featuring the 'fast' D-current exhibit high static and low dynamic excitability, while those with the 'slow' D-current exhibit low static spike counts and high dynamic spike counts.

The D-current, having both activation and inactivation, was manipulated in a slightly different manner. Instead of shifting the midpoint of activation alone, we introduced a concurrent shift of the steady-state inactivation curve, hence moving them synchronously. This allowed us to keep the overlap of the steady-state activation and inactivation curves at a fixed value. The trajectories of 25 selected representative stuttering model neurons are shown in Fig 6C. Again, we find a field of nearly parallel trajectories that mostly start in locations close to the origin of the map (low excitability) and extend toward the upper right corner (higher excitability). Shifting the activation and inactivation midpoint parameters toward more depolarized values, therefore, reduces the impact of the D-current and allows the generation of more intense firing responses under both static and dynamic inputs.

Finally, we find the most intriguing effects when manipulating the kinetics of the D-current. We shift the activation and inactivation time constants in a way that their ratio remains

constant. In effect, we simulate an overall slowdown of the D-current by increasing the maximal activation time constant from 2 ms to 18 ms and proportionally increasing the inactivation time constant, too. Models with the 'fast' D-current tend to produce responses featuring high static and low dynamic total spike counts, while the ones with the 'slow' D-current do the opposite. Visually, this results in trajectories that move across the 2D map (Fig 6D) rather than lining up in parallel with the main diagonal. Inspecting the voltage responses of the stuttering model neuron when 2 ms used for its maximal activation time constant (Fig 6D$_1$) we find a firing pattern that is not unlike that of the delayed firing model neurons. There is a characteristic ramp preceding the first spike, but the firing is fairly regular after that. This behavior is frequently observed in fast spiking neocortical or striatal neurons [20,23,26]. On the other hand, when the time constant is increased, the model tends to emit a single spike with short latency near rheobase and stuttering response at more positive currents. Hence, the kinetics of the D-current has a very important regulatory role in setting the intensity of the firing output under both types of stimulus conditions. Remarkably, gradual changes in the time constants shift the static vs. dynamic firing responses in opposite directions, i.e. the reduction of firing under step currents is complemented by the increase of the synaptic response of the same neuron. This effect is markedly different from those found under the up- or downregulation of the total current or when manipulating their voltage-dependence.

## Discussion

We report here an experimental observation of surprisingly weak relationship between conventional measures of intrinsic excitability and the intensity of neuronal firing under realistic synaptic inputs. Hippocampal neurons expressing either Kv1 channel mediated D-currents or inward rectifying K-currents respond in a divergent manner under the action of static vs. dynamic stimuli indicating correspondingly differentia effects when these two intrinsic currents are subjects of adaptive up- or downregulation. Results from our model simulations verify and aid an explanation of these data. Our results also highlight the importance of using physiologically realistic stimulus protocols when assessing the functional impact of changes in the magnitude of specific intrinsic membrane currents.

### Plasticity of intrinsic excitability and impact on synaptic integration

The variety and intricacy of activity-dependent neuroadaptations in the nervous system are truly remarkable. One main form of plasticity in central neurons is the long-term potentiation of their intrinsic excitability (LTP-IE) that involves activity-dependent regulation of the magnitude of specific voltage-gated membrane currents [1,9,30,31]. As an important example of such adaptations, the dendrotoxin-sensitive K-current, mediated by Kv1 channels in neocortical [23] and striatal [20,32] neurons, is downregulated by LTP protocols traditionally used to induce synaptic modifications [3,29,33]. The resulting changes of neuronal firing responses are well documented in a variety of neuron types but it is less clear how such cellular adaptations translate to the neurons firing properties under natural conditions. Indeed, virtually all reports on LTP-IE demonstrate the changes in intrinsic excitability based upon input-output relationships recorded under current step protocols. EPSP-spike coupling, as a physiologically more relevant phenomenon has been also shown to be affected by LTP-IE [3,6] and these observations clearly motivate further analysis that address the regulation of integrative properties and circuit interactions under intrinsic neuroadaptations.

Our earlier findings on differential regulation of firing responses under plastic changes or manipulations of the intrinsic cellular properties of extended amygdala [24,34] and cultured hippocampal neurons [25] showed that conventional analysis of intrinsic excitability might be

rather limited in respect to estimating the impact of LTP-IE on the integration of complex synaptic inputs. As we have shown, static vs. dynamic firing responses are differentially regulated by voltage-gated currents [24], and the same currents also facilitate a kind of spiking resonance manifesting as frequency-dependent integration of synaptic inputs [25]. Our prior experiments were based on manipulation of the biophysical properties of neurons via dynamic clamp insertion of synthetic Hodgkin-Huxley type conductances [24,25]. It was therefore warranted to demonstrate the effects of pharmacological manipulation of real biological membrane currents on the firing responses. To our knowledge, this approach, together with the cell-type specific analysis of static excitability and synaptic integration, has not yet been used. Application of specific blockers of voltage-dependent currents mimics, at least partially, the neuroadaptive changes observed under activity-dependent downregulation of currents under LTP-IE [3,4,9]. Such pharmacological manipulations are therefore considered as valuable tools to assess functional changes induced by LTP-IE or homeostatic adaptations. While in the present study we did not aim at inducing activity-dependent changes in the intrinsic properties of neurons, our experimental and computational findings are still applicable to assess functional effects of such adaptations. The strength of voltage-dependent currents can be arbitrarily varied in model neurons that better replicates the gradual changes associated with LTP-IE [24,25].

Conveniently, the magnitude of specific voltage-gated membrane currents varies among nerve cells of physiologically diverse cultured hippocampal cultures. Thus, sampling of a high number of such neurons can reveal the impact of variable expression of membrane currents on the firing responses. This is possible mainly because strong correlations exist between the magnitude of specific voltage-activated currents and physiological parameters extracted from the current step responses (e.g. voltage sag vs. $I_h$, inward rectification vs. $I_{Kir}$) [19]. Therefore, the variability of physiological parameters across the population of recorded cells reflects their underlying biophysical diversity [17,35–37]. We are aware of the fact that dissociated hippocampal cell cultures inherently contain heterogeneous populations of multiple neuronal types (e.g., excitatory pyramidal neurons, granule cells as well as inhibitory interneurons). Although different classes of hippocampal neurons can be identified using a relatively simple immunocytochemistry [38], the ratio of certain cell types can vary greatly upon preparation methods or culture conditions [39]. Indeed, in our preliminary immunocytochemical analysis we identified a mixture of putative CA1 and CA3 excitatory neurons as well as GABAergic neurons in our cultures. Furthermore, the overall high degree of physiological diversity is likely associated with the undirected formation of synaptic connections among these cells that are randomly distributed on the coverslip surface. However, we use the physiological and biophysical diversity of these neurons for our benefit, because it allows the correlative analysis of excitability profiles and integrative properties in a high-throughput manner.

The physiological phenotypes we identified in our cell cultures and the corresponding computational models can be considered as representations of type 1 and 2 excitabilities [40]. In particular, regular and delayed firing type cells increase their firing rate in a smooth, fairly continuous fashion and they typically emit only one action potential when the current level reaches rheobase. In contrast, stuttering type neurons exhibit far more irregular firing output and their spike count vs. current (I-O) relationship shows sudden jumps. While it is not common to find neurons that initiate firing with multiple spikes at rheobase levels, they often start with a single action potential and then suddenly increase their firing rate as the stimulus current increases. In this respect they show behavior similar to that observed in fast spiking interneurons [40]. Kv1 channel mediated D-currents appear to play an important role in the expression of stuttering responses [20] and our experiments with 4-AP application further supported this notion. In an elegant dynamic clamp study the authors manipulated the

biophysical properties of pyramidal neurons to change the type of their excitability [41]. Here, addition of a computer-synthesized Kv-current shifted the firing responses of pyramidal neurons from regular (type 1) to stuttering (type 2) mode. The single spiking response near rheobase and the shortened spike latency were remarkably similar to those we observed in our experiments and computational models that targeted the D-current.

Due to their physiological heterogeneity, we found a great variability of voltage responses and firing patterns of hippocampal neurons under static vs. dynamic stimulation. More importantly, correlation between measures of excitability associated with the two stimulus protocols was surprisingly low. In essence, these experiments suggest that estimating the intensity of postsynaptic firing responses by analyzing only the static I-O functions of the same neurons is not viable. Of course, one can readily accept this limitation considering the different temporal structure of the experimentally applied DC waveforms and the complex synaptic currents occurring in natural conditions. However, it can be still argued that comparing the static firing responses of two neurons or the same one in two treatment conditions can be informative in estimating which neuron would fire more intensely under the action of identical synaptic inputs. As we found, even this is assumption is not applicable, because the sensitivity of neurons to static vs. fluctuating current inputs can be vastly different depending on their biophysical properties. As an example, one can detect a major change in the current step responses following LTP-IE that would suggest accordingly significant alterations in the synaptic responses of the same neuron. The LTP-induced downregulation of the D-current in hippocampal parvalbumin-expressing basket cells has been shown to induce specific changes in their static firing responses including the leftward shift of their I-O functions and the marked decrease of the first spike latency [3]. These findings suggest the concurrent upregulation of the synaptic responses of basket cells and their facilitated recruitment in network activity at gamma frequencies. Nevertheless, our present data suggest a more moderate impact of the D-current in regulating synaptic integration than the firing under step current stimulation.

This latter voltage-activated current appears to be especially interesting, because slight changes in its kinetic properties will dramatically reshape the firing output of the neurons under both static and dynamic inputs. Depending on the activation/inactivation time constants of this current, single spiking/stuttering behavior [20] or the prolonged first spike latency can be also observed [23,29,32,42], the latter being a characteristic feature of fast-spiking neurons. The variety of observable firing patterns under the manipulation of the Kv1 channel-mediated currents and the underlying changes in intrinsic excitability clearly serve as motivation for electrophysiologists to focus on the impact of this current in regulating synaptic integration and long-term plasticity.

Additionally, our data here show that the impact of up- or downregulation of the D-current can be somewhat counterintuitive when the neurons are operating under the action of fluctuating synaptic inputs. Increasing the magnitude of the D-current has a stronger effect on current step responses than on firing under synaptic inputs. The effect of changing the activation and inactivation speed of this current has an even more contrasting effect on those. Accordingly, the D-current displays some unique properties with respect to regulating the firing output of neurons and they certainly justify further investigations.

The hyperpolarization-activated cation current $I_h$ has been also extensively investigated in respect to its role in regulating intrinsic excitability and dendritic integration. This current is expressed both in the regular firing and stuttering type hippocampal neurons, and it is reasonable to assume that $I_h$ also regulates static vs. dynamic firing responses in a differential manner. Our earlier computational studies suggested that upregulation of the h-current can actually increase intrinsic excitability and postsynaptic firing responses due to its depolarizing effect on the resting membrane potential [24,25]. Similar excitability increasing effect has been also

verified experimentally in dendrites of CA1 pyramidal neurons after epilepsy-related upregulation of their h-current [43]. The dual effect of $I_h$ on membrane resistance [44,45] and resting membrane potential makes it therefore more challenging to accurately assess its net effect on firing output. In our present biological experiments, we did not address the net effect of $I_h$ on firing responses, however, in a separate study, we observed a clear hyperpolarizing shift of the resting membrane potential after pharmacological blocking of this current [46]. Additionally, we performed model simulations to examine the effect of $I_h$ on the physiological properties and firing responses of stuttering type neurons. Interestingly, removal of $I_h$ in such model neurons had a minor effect on the cumulative spike counts under both current step stimulation and simulated synaptic bombardment (see S1 Fig). Also, spike latencies were barely affected by the removal of $I_h$ indicating that this parameter is mainly regulated by the strong D-currents in stuttering cells. Indeed, additionally removing the D-current in such model neurons largely reproduced the changes demonstrated in Fig 5D–5F.

## The rationale of current step stimulation and simulated synaptic inputs

The differential regulation of firing responses observed in our dynamic clamp experiments and computational models suggests that these effects are closely related to the voltage dependence and kinetics of the involved membrane currents. However, one can rightfully ask whether random variations in the passive membrane properties can, at least partially, account for the great diversity of firing responses observed in our data. This problem is further justified by acknowledging that simplistic phenomenological models can predict the firing patterns of biological neurons in a surprisingly accurate manner [47]. Hence, we performed simulations with leaky integrate-and-fire model neurons where 4 parameters were randomly varied, similarly to that we did with the biophysically more realistic models (see S2 Fig). The membrane capacitance, reversal potential and maximal conductance of the leakage current, and the voltage threshold of spiking were varied across 200 implementations. While cumulative spike numbers from such model runs varied in a wide range (5-fold), the correlation between the 2 parameters remained very tight ($Corr_S = 0.99$). Clearly, random variations in the passive membrane properties of neurons cannot explain the very significant disparity between static responses and spiking driven by synaptic inputs. Therefore, differential regulation of firing output is found only in biophysically more realistic computational models that well reproduce the behavior of real biological neurons under simulated synaptic bombardment.

Considering the above findings, we conclude that parameters of static excitability, as measured in current step protocols, yield limited value in estimating the firing activity of neurons as they integrate real synaptic inputs. This is reasonable because the membrane potential of postsynaptic neurons fluctuates widely under synaptic activation but evolves in a more restricted regime when constant stimulus current is used. Indeed, the membrane potential between spikes does not visit levels below resting membrane potential when depolarizing current steps are used in patch clamp settings. In normal conditions, however, neurons receive a mix of excitatory and inhibitory synaptic conductances allowing transient hyperpolarizations of the membrane potential between action potentials (potentially reaching the reversal potential of GABA-gated currents). In such conditions, the relative contribution of low-threshold ($I_{CaT}$) or hyperpolarization-activated ($I_h$, $I_{Kir}$) voltage-gated currents in shaping the firing output becomes stronger than under static current injection.

Our present electrophysiological study highlights the value of simulated synaptic inputs in the analysis of functional changes facilitated by plasticity of intrinsic properties, and, in a more general sense, when voltage-activated membrane currents are subjects of neuromodulation or long-term neuroadaptations. One key technical benefit of such a technique is that biological

neurons can be exposed to accurately controlled and replayed synaptic conductance waveforms [15,48], either previously recorded in vivo, or synthesized by computer. The dynamic clamp can reproduce many of the important temporal features of natural synaptic activation including the random variations in the amplitude and kinetics of the postsynaptic currents and their temporal patterns (e.g. oscillatory network activity) [25,49]. Importantly, temporally complex synthetic synaptic inputs delivered by dynamic clamp elicit firing responses that are far more reliable and precise than those acquired under repeated constant current step stimulation. While the early pioneering work of Mainen and Sejnowski has clearly demonstrated the great value of fluctuating current stimuli in probing neuronal response properties [50], more systematic investigations with mixed excitatory and inhibitory synaptic inputs via dynamic clamp revealed spike responses with submillisecond temporal jitter [51,52]. Dynamic clamp remains as a very effective tool to investigate synchronization [53], information processing [48,54] and resonant properties [25] in biological neurons receiving various types of synaptic inputs. Indeed, the use of dynamic clamp in investigations of cellular properties and circuit interactions has been growing steadily [55–57] due to its power in revealing effects that might remain masked using more conventional tools of patch clamp electrophysiology. The differential impact of specific voltage-gated currents in regulating the firing output, as shown by our present data, further reinforces the value of such technology and hopefully motivates electrophysiologists to routinely include such protocols in their investigations.

## Methods

### Cell cultures and electrophysiology

Our electrophysiological experiments were performed on cultured mouse hippocampal neurons. We prepared primary cultures of embryonic hippocampal cells from CD1 mice [58,59]. The cells were seeded onto poly-L-lysine and laminin-coated glass coverslips in 24-well plates. Neurobasal medium (Invitrogen) containing 2% B27 supplement (Invitrogen), 0.5 mM Glutamax (Gibco) and 5% FCS (Invitrogen) was used for plating and for a complete medium change on the first day after plating (DIV1). On the 5th, 9th and 12th day after plating, half of the culture medium was changed to BrainPhys medium containing SM1 supplement. To inhibit glial cell division, 10 μM cytosine β-D-arabinofuranoside (Sigma-Adrich) was added to the cultures between DIV4 to 6. Recordings were performed 14–17 days after plating when the neurons typically exhibited robust spontaneous bursting activity, mature cellular properties and synaptic connections. The experiments were performed in whole-cell patch clamp conditions at room temperature (22–24°C) using a MultiClamp 700B amplifier (Molecular Devices). The composition of the extracellular solution (artificial cerebrospinal fluid, ACSF) was (in mM): NaCl 140, KCl 5, $CaCl_2$ 2, $MgCl_2$ 1, HEPES 5, D-glucose 10; pH 7.45. The patch electrodes were pulled from borosilicate glass, had 6–8 MOhm resistance and filled with the following solution (in mM): K-gluconate 100, KCl 10, KOH 10, $MgCl_2$ 2, NaCl 2, HEPES 10, EGTA 0.2, D-glucose 5; the pH was set to 7.3. Access resistance (max. 22 MOhm accepted) was carefully compensated to gain accurate membrane potential readings during current injection and in dynamic clamp experiments. The input current waveforms and voltage responses were sampled at 20 kHz and low-pass filtered at 6 kHz in DASYLab v. 11. (National Instruments).

### Current step stimulation and static excitability

Each neuron in this study was exposed to two types of intracellular stimulation. The first protocol was designed to gain an accurate estimation of the cells' many physiological parameters and measures of their intrinsic excitability. Here, we stimulated the neurons using step current injections of 400 ms duration, starting at -160 pA and incremented by 5 pA until the strongest

depolarizing current of +200 pA was reached. The cycle period of this stimulation was 1.25 s. The input-output curve of the neuron was obtained by counting spikes for each current level up to +200 pA. The total (cumulative) spike count for each stimulated neuron was also calculated in such experiments. The rheobase and the total spike count served as the two key parameters we used to characterize the neurons intrinsic excitability in such conditions (static excitability). In addition to measures of excitability we extracted multiple physiological parameters such as membrane resistance, voltage sag, afterdepolarization, estimated membrane capacitance [19]. Subjective classification of the neurons into 3 main groups was based on initial visual inspection of the current step responses and the analysis of the physiological parameters. We used our custom software NeuroExpress to extract a total of 38 physiological parameters from the voltage traces [19]. In addition to the standard parameters, we introduced the Latency$_{1.3}$ parameter to characterize the spike response speed of the neuron under static stimulation. This was done by fitting the recorded first spike latencies with a Belehradek function (e.g. Fig 1E, 1G and 1I) and reading the function's value at 1.3-times the observed rheobase of the neuron.

## Simulated synaptic inputs and dynamic clamp

Additional excitability parameters of the neurons were obtained in dynamic clamp conditions using simulated synaptic inputs as stimulation [24,25]. In such experiments the firing of the biological neurons was evoked by exposing them to gradually strengthening levels of simulated postsynaptic currents as described previously [24,60]. Briefly, we generated two independent voltage waveforms mimicking the firing activity of virtual presynaptic excitatory and inhibitory neurons firing in a Poissonian pattern. The two waveforms were fed to the dynamic clamp system to generate mixed excitatory (AMPA-) and inhibitory (GABA-type) synaptic currents. The time constant of the simulated synaptic input was 10 ms for both the AMPA- and GABA-connections, while their reversal potential was set to 0 mV and -72 mV, respectively. The strength of the excitatory synaptic conductance was gradually increased from 0 to 25 nS in increments of 0.5 nS, while the inhibitory conductance was incremented by 1 nS in the successive sweeps of stimulation. One sweep of this protocol consisted of the delivery of the mixed excitatory/inhibitory synaptic input for 2.5 s followed by a rest for an additional 2.5 s yielding a cycle duration of 5 s. For the dynamic clamp experiments we used the StdpC v. 2012 platform that allowed us the automatic incrementing of synaptic conductance parameters (via scripting) in the successive sweeps of stimulation [60].

## Computational models and virtual electrophysiology

In the computer simulations we exposed model neurons to the 2 types of stimulation that accurately matched those we used for the biological neurons. The model neurons' responses were then analyzed using the algorithms we employed for the electrophysiological data. We designed three types of model neurons aiming to reproduce the physiological properties of the biological neuron phenotypes and their dynamical responses. These were 3-compartmental model neurons consisting of a somatic, dendritic and axonic compartment. Passive membrane properties and parameters of the voltage-dependent currents are listed in S1 and S2 Tables, respectively. All intrinsic voltage-dependent currents were implemented as standard Hodgkin-Huxley types:

$$I_i = g_i m_i^p h_i (E_i - V),$$

The total maximal conductances ($g_i$) of the currents for each model instance were drawn from a Gaussian-distribution to simulate a biophysically diverse neuron population. Mean

maximal conductances and their standard deviations are presented in S3 Table. The reversal potential of the currents ($E_i$) and their other kinetic parameters were not varied across implementations. Two-hundred model instances were simulated for each biophysical phenotype. Differential equations for the activation ($m$) and inactivation ($h$) shared the same form ($x$ being either $m$ or $h$):

$$\frac{dx}{dt} = \frac{x_\infty(V) - x}{\tau_x(V)},$$

where voltage-dependent steady-state activation and inactivation were described by sigmoids:

$$x_\infty(V) = \frac{1}{2} + \frac{1}{2}\tanh\left(\frac{V - V_{x,1/2}}{V_{x,sl}}\right).$$

Here, $V_{x,1/2}$ denotes the midpoint of the activation/inactivation sigmoid and $V_{x,sl}$ is its slope. Time constant of the activation and inactivation were bell-shaped functions of the membrane potential:

$$\tau_x(V) = \left(\tau_{x,max} - \tau_{x,min}\right)\left[1 - \tanh\left(\frac{V - V_{\tau x,1/2}}{V_{\tau x,sl}}\right)^2\right] + \tau_{x,min}$$

Here, $\tau_{x,max}$ and $\tau_{x,min}$ indicate the maximal and minimal values of the bell-shaped functions, $V_{\tau x,1/2}$ indicates their midpoint and $V_{\tau x,sl}$ sets the slope of the functions. The Ca-dependent K-current and internal Ca-dynamics were based on the formalism in [61]. Synaptic currents were described using a first-order kinetics of transmitter release [62] as:

$$I_{syn} = g_{syn}S(E_{syn} - V),$$

where $S$ is the instantaneous synaptic activation term yielding the following differential equation:

$$\frac{dS}{dt} = \frac{S_\infty(V_{pre}) - S}{\tau_{syn}(1 - S_\infty(V_{pre}))}$$

The steady-date synaptic activation term depends on the presynaptic membrane potential as

$$S_\infty\left(V_{pre}\right) = \tanh\left(\frac{V_{pre} - V_{th}}{V_{slope}}\right),$$

when $V_{pre} > V_{th}$, otherwise $S_\infty(V_{pre}) = 0$. $V_{pre}$ denotes the presynaptic membrane potential waveform that is stored in ASCII files and identical to those used in the dynamic clamp experiments. The reversal potential of the excitatory and inhibitory synaptic connections was 0 and -72 mV, respectively.

## Supporting information

**S1 Table. Passive membrane parameters of the three neuron models.** $g_{sx}$ is the electrical coupling conductance between the soma and axon compartments. $g_{sd}$ indicates the coupling between the soma and dendrite.
(DOCX)

**S2 Table. Parameters of the voltage-dependent currents for the three neuronal phenotypes (R: regular firing, D: delayed firing, S: stuttering).** So, Ax and De indicate the percentage of conductance allocated for the somatic, axonic and dendritic compartments.
(DOCX)

**S3 Table. Parameters used to create biophysically diverse model neurons.** Maximal conductances of voltage-gated currents, membrane capacitance, leakage reversal potential, and coupling conductances between compartments were varied in a Gaussian distribution. The values are expressed as mean ± S.D. of the distributions.
(DOCX)

**S1 Fig. Effects of removal of the D-current from the stuttering type model neurons without $I_h$.** (A) Spike latency parameters are scatter plotted against the membrane resistance. Filled symbols represents the models with intrinsic D-current, the open ones correspond to those after the removal of ID. (B) Scatter plot of cumulative spike counts from the same set of model neurons (synaptic vs. current step responses). (C) Pooled cumulative spike counts shift in a differential manner when static vs. dynamic inputs are used. The data demonstrate a minor effect of $I_h$ on firing output relative to that of the D-current.
(TIF)

**S2 Fig. Parameters of intrinsic excitability are tightly correlated under the variation of passive membrane properties of a leaky integrate-and-fire model.** (A) The threshold AMPA-conductance is plotted against the rheobase of 200 LIF model instances. (B) Dynamic vs. static cumulative spike counts are plotted for the same model instances.
(TIF)

## Author Contributions

**Conceptualization:** Attila Szücs.

**Formal analysis:** Adrienn Szabó, Attila Szücs.

**Funding acquisition:** Katalin Schlett, Attila Szücs.

**Investigation:** Adrienn Szabó, Attila Szücs.

**Methodology:** Attila Szücs.

**Software:** Attila Szücs.

**Writing – original draft:** Adrienn Szabó, Katalin Schlett, Attila Szücs.

**Writing – review & editing:** Adrienn Szabó, Katalin Schlett, Attila Szücs.

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
