## [Decision Letter · Decision Letter 0]

29 Apr 2021

Dear Dr. Szücs,

Thank you very much for submitting your manuscript "Conventional Measures of Intrinsic Excitability Are Poor Estimators of Neuronal Activity Under Realistic Synaptic Inputs" for consideration at PLOS Computational Biology.

As with all papers reviewed by the journal, your manuscript was reviewed by members of the editorial board and by several independent reviewers. In light of the reviews (below this email), we would like to invite the resubmission of a significantly-revised version that takes into account the reviewers' comments.

We cannot make any decision about publication until we have seen the revised manuscript and your response to the reviewers' comments. Your revised manuscript is also likely to be sent to reviewers for further evaluation.

Sincerely,

Joseph Ayers, PhD

Associate Editor

PLOS Computational Biology

Lyle Graham

Deputy Editor

PLOS Computational Biology

Reviewer's Responses to Questions

**Comments to the Authors:**

Reviewer #1: The authors conducted a thorough analysis of the roles of Kv1 and Kir K+ currents in excitability of neurons by comparing electrical responses of cultured mouse hippocampal neurons to standard long square current protocols and to injected fast artificial synaptic inputs via dynamic clamp. They found that there are at least three types of neurons separated by their response to long pulses of current. Two of these types exhibited opposite effects of these stimulation protocols. They found that the responses of neurons classified as delayed firing type, could be explained by relatively upregulated KIR current; and the responses of neurons described as being of stuttering type are explained by relative upregulation of Kv1 channels (D-type K current). These study cleverly compares results of electrophysiological measurements, pharmacological blockade of the key currents, dynamic clamp stimulation, and computational modeling. The study is clearly written and provides interesting and general results. If questions and issues described below are properly addressed this study could have high impact in the field of neuroscience.

Major

I suggest that these classifications and types of responses should also acknowledge the role played by hyperpolarization activated (h-) current in the stuttering neurons. Figure 1 B suggests that neurons with the delayed firing practically have no h-current while stuttering neurons and regular firing neurons have prominent sag potential reporting notable presence of h-current. In the model this distinction is less obvious.

It appears that the role of h-current contributed to a counter-intuitive change in responses after blockade of Kv1 channels.

314 “Somewhat paradoxically, the latency parameter increased after D-current removal in the stuttering models (Fig. 5D) indicating that these cells start firing with longer

latencies but at lower step current levels compared to those having their intrinsic D

currents intact.” Could this interesting phenomenon be explained by partial deactivation of h-current? This could be easily tested in the model and would serve as strong validation of the model and shed light on the role of h-current.

These two types of neurons appear to represent type I and type III excitability. It would help this study to discuss presented results in the context of types of excitability.

The applicability of the term stuttering is not obvious for me and should be explained and well defined as a set of clear formal criteria for classification. It appears that the authors determine this type of neurons by the predominantly silent interval after initial response of neurons to long square current pulse. Are these neurons bursting? How many spikes are in the groups of spikes? Two? Are there more spikes in the first bursts/groups? Could these bursts be due to reminiscent activated h-current?

Relation of the results of this study to the “activity-dependent regulation of intrinsic excitability” should be better explained.

Minor

Figure 1 A, B, and C should have same voltage scale bar.

Are neurons with rest membrane potentials above -50 mV excitability blocked?

Here and below, the pictogram of experimental recording in the top right corner is blurred and not contrast. It should be improved.

186: “threshold AMPA-conductance as a function of the threshold current level (rheobase) for” the term rheobase could be introduced as the threshold current level earlier in the text.

Experimental rest potentials are much more depolarized than model rest potentials. Could the authors discuss this discrepancy.

Figure 3C it is not clear whether the neurons spiking at the end of stimulation produced spikes in the beginning. Probably, if the traces are plotted in different colors then the traces could be distinguished.

Figure 3I does not show data and thus it could not match experimental data.

Figure 3 J visibly exhibits a trend different from experimental.

227 “For each model phenotype we generated 200 instances and subjected those to the two types of stimulation as used with the biological neurons. “

How did you produce these 200 instances? Did the model phenotype appear as an emergent property? If not, what constraints did you use to obtain them?

233 Is the model multicompartmental?

What would 4ap application do to the delayed neurons? What would application of BaCl2 do to stuttering neurons? The data should be shown:” It is also noteworthy that the actions of the two channel blockers were cell type specific, meaning that BaCl2 was found to be ineffective when applied on stuttering type neurons while 4-AP was ineffective on delayed firing type ones (data not shown). These observations indicate that Kir and Kv1 channels are expressed in separate populations of cultured hippocampal neurons.

”

Why figure 4 is not reproduced in the model?

Why model analysis in Figure 5 does not have a corresponding experimental figure?

Reviewer #2: This manuscript combines electrophysiology and dynamic clamp experiments on cultured rat hippocampal neurons with computational ensemble modeling. The major conclusion drawn by the authors is that the voltage responses elicited from neurons by traditional current step stimuli are a poor predictor of the effect of more realistic stimuli (in the form of barrages of excitatory and inhibitory synaptic inputs) on neuronal firing rates and other excitability measures. Two potassium membrane currents are identified as major determinants of physiologically meaningful excitability measures. Interestingly, the effects of these membrane currents can be in the opposite direction for realistic stimulation compared to step current stimulation, and can differ between neuron types (the authors distinguish three types based on the temporal spiking profile and the presence or absence of sag currents).

The experiments and models are well thought out and executed. In particular, the use of ensemble modeling to replicate biological variability across populations of neurons is highly appropriate. The conclusions are well founded in the experimental and modeling results, the figures are clear, and the text is concise and to the point.

I have only one comment that I hope the authors can address: the literature cited could more thoroughly relate the results presented here to previous work in the literature. Most notably, it is surprising that the seminal work of Mainen and Sejnowski on the poor suitability of current step stimuli for the characterization of neocortical response dynamics is neither cited nor discussed here: Mainen ZF & Sejnowski TJ (1995). Science 268: 1503-1505.

**Have all data underlying the figures and results presented in the manuscript been provided?**

Reviewer #1: None

PLOS authors have the option to publish the peer review history of their article (what does this mean?). If published, this will include your full peer review and any attached files.

Reviewer #1: No

Reviewer #2: No

**Have the authors made all data and (if applicable) computational code underlying the findings in their manuscript fully available?**

Reviewer #2: **No: **The authors state that "All model simulation data files will be made available from the time of publication at the Figshare data repository."

Model simulation data comprise only a fraction of the data presented in the manuscript. Experimental and dynamic clamp data as well as simulation and dynamic clamp code should also be made available, as per journal policy.
---

## [Decision Letter · Decision Letter 1]

24 Aug 2021

Dear Dr. Szücs,

We are pleased to inform you that your manuscript 'Conventional Measures of Intrinsic Excitability Are Poor Estimators of Neuronal Activity Under Realistic Synaptic Inputs' has been provisionally accepted for publication in PLOS Computational Biology.

Best regards,

Joseph Ayers, PhD

Associate Editor

PLOS Computational Biology

Lyle Graham

Deputy Editor

PLOS Computational Biology

Reviewer's Responses to Questions

**Comments to the Authors:**

Reviewer #1: The authors addressed my concerns and I support acceptance of this article.

Reviewer #2: In this revised manuscript, the authors have adequately addressed my minor comment, and the more extensive comments of the other reviewer. This already well written manuscript on a carefully designed and executed dynamic clamp study is now further improved, and should be of interest to the readership of PLoS Computational Biology.

**Have the authors made all data and (if applicable) computational code underlying the findings in their manuscript fully available?**

Reviewer #1: None

Reviewer #2: Yes

PLOS authors have the option to publish the peer review history of their article (what does this mean?). If published, this will include your full peer review and any attached files.

Reviewer #1: No

Reviewer #2: No

---

## [Editor Report · Acceptance letter]

9 Sep 2021

PCOMPBIOL-D-21-00383R1 

Conventional Measures of Intrinsic Excitability Are Poor Estimators of Neuronal Activity Under Realistic Synaptic Inputs

Dear Dr Szücs,

I am pleased to inform you that your manuscript has been formally accepted for publication in PLOS Computational Biology. Your manuscript is now with our production department and you will be notified of the publication date in due course.

With kind regards,

Andrea Szabo
